# Infrared and Visible Image Fusion via Attention-Based Adaptive Feature Fusion

**DOI:** 10.3390/e25030407

**Published:** 2023-02-23

**Authors:** Lei Wang, Ziming Hu, Quan Kong, Qian Qi, Qing Liao

**Affiliations:** 1Hubei Key Laboratory of Optical Information and Pattern Recognition, Wuhan Institute of Technology, Wuhan 430205, China; 2School of Art & Design, Wuhan Institute of Technology, Wuhan 430205, China

**Keywords:** image fusion, adaptive fusion strategy, attention mechanism

## Abstract

Infrared and visible image fusion methods based on feature decomposition are able to generate good fused images. However, most of them employ manually designed simple feature fusion strategies in the reconstruction stage, such as addition or concatenation fusion strategies. These strategies do not pay attention to the relative importance between different features and thus may suffer from issues such as low-contrast, blurring results or information loss. To address this problem, we designed an adaptive fusion network to synthesize decoupled common structural features and distinct modal features under an attention-based adaptive fusion (AAF) strategy. The AAF module adaptively computes different weights assigned to different features according to their relative importance. Moreover, the structural features from different sources are also synthesized under the AAF strategy before reconstruction, to provide a more entire structure information. More important features are thus paid more attention to automatically and advantageous information contained in these features manifests itself more reasonably in the final fused images. Experiments on several datasets demonstrated an obvious improvement of image fusion quality using our method.

## 1. Introduction

Infrared and visible image fusion (IVIF) are widely used in many fields such as pattern recognition [1], remote sensing [2,3,4,5,6], video surveillance [7], and modern military missions [8]. Infrared images and visible images have different imaging properties. Infrared images contain more information about thermal radiation and mainly use pixel intensity to distinguish a significant target from the background. Visible images contain more information about texture structure and mainly use gradients to reveal rich detail information about the scene. Therefore, the goal of image fusion is to retain as much advantageous features of source images as possible, such as texture information in visible images and salient target information in infrared images.

Fusion algorithms are usually classified into traditional methods and deep learning-based methods. Traditional methods usually include three approaches: multi-scale-based methods [9,10,11]; sparse representation-based [12,13,14] and low-rank representation-based [15,16] methods. Multiscale methods usually decompose source images into different scales for feature extraction and use an appropriate fusion strategy to fuse features on each scale. The inverse operator is then used to reconstruct the fused image. Their performance highly depends on multiscale feature extraction algorithms. In sparse representation (SR)-based methods, the same dictionary is used to represent both source images. In low-rank representation (LRR)-based methods, the same coefficient matrix is used to extract the significant components from all source images. For SR and LRR-based methods, it is difficult to handle complex source images due to the long dictionary learning period. Traditional methods have poor feature extraction capability and mostly use element-by-element maximum fusion strategy or element-by-element weighted fusion strategy. The fusion performance of these methods is limited because the manually designed fusion rules cannot fuse features adaptively.

Deep learning (DL) methods use deep neural networks to extract features and reconstruct fused images, overcoming the shortcomings of feature extraction capability of traditional methods. Considering each source image pair is generated under the same scene but sensor specific imaging mechanisms, Xu et al. [17] first proposed a fusion method (DRF) for decoupled representation. Xu et al. [18] proposed a double encoder-decoder method (CUFD) based on common and unique features. One encoder-decoder network focuses on extracting shallow and deep features, and the other one is used to decompose source images into similar common parts and unique parts according to pixel intensity distribution in source images. Fu et al. [19] proposed a two-branch network for decoupling features. The features are extracted and decoupled by an encoder including a detail branch for dense connection and a semantic branch for fast downsampling, and then fused under an addition strategy. Li et al. [20] proposed deep image decomposition fusion (DIDFuse) method, which uses a pair of encoder-decoder networks with shared parameters to decompose source images into two types of features: similar background features and dissimilar detail features. The features of the same type from different sources are first synthesized with an addition strategy, then the features of different types are cascaded and fed into the decoder. Kong et al. [21] treated IVIF as a normalized modulation process and proposed a pixel-adaptive normalization (PEAN) mechanism to fuse the summed structure and the summed detail feature, obviously improving the image fusion quality. However, the features of the sample type from different sources still employ addition fusion strategy.

Although extensive methods have been trying to decompose source images into source-invariant structure features of the scene and source-specific modal features, the extracted structure features from different sources still differ to a large extent due to their vast discrepancy in imaging mechanisms. The above mentioned DL methods focus only on complementary source-specific modal features, ignoring the complementarity between source-invariant structure features. Moreover, most of these DL methods use hand-designed simple fusion strategies for synthesizing the features of the same type but from different sources, such as average addition or addition with fixed weights designed by hand. In fusion of features of different types, simple concatenation strategy is the most choice. These simple strategies only provide fixed and manually designed weights for different features, not taking the relative significance of different features into account and thus limiting the fusion performance.

To address these issues, this paper proposes an adaptive fusion network for IVIF tasks, based on multi-scale channel attention mechanism [22]. Source images are first decomposed into source-invariant common structure features and source-specific modal features. An attention-based adaptive fusion (AAF) module within this network learns the importance of extracted features and thus can dynamically assign different weights to different features in fusion process, according to their relative importance. The contributions of this paper are summarized as follows:1.To the best of our knowledge, it is the first time an adaptive fusion strategy has been introduced in IVIF. The AAF module based on multiscale channel attention mechanism dynamically generates fusion weights for different features along the channel dimension in a pixel-wise dynamic weighting manner;2.In addition to applying the AAF strategy to synthesize structural features and modal features, we also apply it to synthesize structural features from different sources before reconstruction, in contrast to usual addition strategy applied to features of the same type in current methods. As a consequence, our synthesized structural features reflect more entire structural information;3.Our network can pay more attention to the dominant features and generate fused images with super-high quality.

The remaining part of this paper is organized as follows. Section 2 introduces related work, including image fusion based on DL, fusion strategies, and attention mechanisms. Section 3 describes the details of attention-based adaptive fusion module and our fusion framework. Then, experimental results are presented and discussed in Section 4. Finally, conclusions are drawn in Section 5.

## 2. Related Work

In this section, we first review some DL-based image fusion methods proposed in the past, and then introduce the existing fusion strategies. Finally, some advanced attention mechanism methods are introduced.

### 2.1. Deep Learning Image Fusion Methods

**AE-based fusion framework.** AE-based frameworks are an important branch of deep learning methods that train autoencoders to achieve feature extraction and feature synthesis. Xu et al. [17] first proposed a fusion method with a decoupling representation (DRF), which decomposes source images into source invariant scene features and source specific attribute features, and fuses the scene features and attribute features before feeding them into a pre-trained generator to reconstruct target images. CUFD [18], Dual-branch Net [19], and DIDFuse [20] also use similar ideas in feature decoupling. However, these methods use simple strategies to fuse the structure and modal features, such as addition and concatenation, which limits the fusion performance because these strategies do not take the relative importance between different features into account. Kong et al. [21] proposed an adaptive normalization mechanism-based fusion method (Normfuse), which injects the detail (modal) features into the structure feature. This adaptive normalization mechanism significantly improves the fustion performance. However, Normfuse still uses addition strategy to fuse features of the same type.

**CNN-based fusion framework.** In recent years, convolutional neural networks (CNNs) have gradually become popular in the field of image fusion. One kind of CNN-based fusion approaches use only pre-trained CNN networks to implement activity level measurements or fusion of extracted features, but the overall fusion framework is still traditional. For example, Li et al. [23] use a deep learning framework based on VGG-19 networks to extract and fuse multilayer features of detail information. However, these CNN models still employ manual design rules for feature fusion. Zhang et al. [24] proposed a CNN-based generic image fusion model (IFCNN), which is composed of three modules: feature extraction layer, feature fusion layer, and image reconstruction layer. It is worth noting that, unlike the previously mentioned CNN approaches, the training of the fusion layer of IFCNN synchronizes with the model. Therefore, IFCNN alleviates the limitations imposed by manually designed fusion rules. Xu et al. [25] proposed an unsupervised end-to-end fusion network (U2Fusion) to avoid the drawbacks of manually set fusion rules. By automatically estimating the relative importance of features through feature extraction and information metrics, the network can be trained to maintain the adaptive similarity between the fused and source images.

**GAN-based fusion framework.** GAN-based models also learns the direct mapping from source image pairs to fused images in an end-to-end manner. Ma et al. first introduced GAN into image fusion task (FusionGAN) to generate fused images via a game playing between discriminators and generators [26]. To improve the quality of detail information and edge sharpening of hot targets, Ma et al. [27] proposed a new end-to-end model on the fusion framework of GAN by designing additional detail loss and target enhancement loss functions. However, a single discriminator may cause the fused image preferring visible or infrared images. Therefore, Ma et al. [28] proposed a dual discriminator conditional generative adversarial network (DDcGAN), which uses two discriminators to balance the similarity distribution between the fused image and the source images. Since GANs-based methods mostly focus on the information of the whole input image, they cannot identify important features. Therefore, Li et al. introduced a multiscale attention mechanism in the GAN-based fusion framework [29] to encourage the generators and discriminators to focus more on the most distinguishing regions. Moreover, Zhou et al. [30] developed a dual-discriminator generative adversarial network (SDDGAN) where an information quantity discrimination (IQD) block was designed to guide the image fusion progress and supervise semantic information of source images in the fused image. Reference [31] designed a unified gradient and intensity-discriminator generative adversarial network for gradient and intensity retention in different image-fusion tasks. However, the end-to-end CNN-based approaches and the GAN-based approaches lack a stage of feature extraction, resulting in poor fusion.

### 2.2. Existing Convergence Strategies

The key to image fusion depends on feature extraction and fusion strategies. However, in the past extensive attention have been paid on designing suitable feature extraction methods, and few studies have been focused on designing fusion strategies. The fusion strategies employed in past fusion methods mainly include addition [15,17,20,21], concatenation [15,18,24,32], average [33], choose-max [11,12,24], max-$l_{1}$ [13], and $l_{1}$-norm [20,23,26]. Among them, the max-$l_{1}$ strategy is used to fuse the sparse coefficients of the source image pair. It generates a binary mask for fusion using the larger sparse coding coefficients by comparing two sparse codes. In [23], feature maps are used to calculate initial activity level maps using $l_{1}$-norm, and then final activity level maps are calculated using the averaging operator. In general, these fusion rules rely on setting values, without estimating the relative importance of extracted features. For example, the features extracted by different methods may correspond to bright regions or dark regions. Choose-max strategy will retain those features corresponding to bright regions and ignore those corresponding to dark regions. However, features corresponding to dark regions may contain valuable information. Therefore, in DL methods, a soft selection fusion strategy [34] with contextual information awareness is proposed. While there are many differences among the various fusion strategies, the differences in implementation details are negligible when abstracting these methods into a mathematical form. The feature fusion formulas for different strategies are shown in Table 1. Linear strategies such as addition and concatenation are not context-aware. In contrast, the soft selection strategy utilizes two input feature maps for guidance and obtain a weight map of the same size as the input features. Moreover, the sum of the weights applied to the two input features must be 1 in the soft selection strategy. This allows the network to dynamically adjust the fusion weights between each pixel of the input features X and Y, while it is not the case with other fusion strategies.

### 2.3. Attention Mechanism

Attention mechanisms have played a very important role in serial and transformation models. They improve the model performance by focusing on the key information wanted in tasks and reducing the attention to unrelated information. Hu et al. first proposed a squeezed incentive network (SENet) based on channel weight assignment [35]. SENet uses a global average pooling layer to compress the input feature map to obtain global features, and then squeezes and expands the channel dimensionality of global features to learn the relative importance of different features. Woo et al. [36] proposed a convolutional block attention module (CBAM) based on channel attention and spatial attention. CBAM adds a spatial attention module to the channel attention module of SENet serially, achieving adaptive feature refinement. Fu et al. [37] proposed a combined spatial attention (SA) and channel attention (CA) mechanism in a parallel dual attention network (DANet), which can capture global dependencies and remote context information more effectively. Liu et al. [38] proposed a pyramidal attention network (PANet) to learn feature relations at different distances in a multiscale feature pyramid. Dai et al. [22] proposed a multiscale channel attention module (MS-CAM), which adds the local scale channel attention to the global scale channel attention, achieving channel attention at both scales. Same as SENet, the global scale channel attention of MS-CAM helps to extract large target features from the global statistical information. The local scale channel attention of MS-CAM is based on SENet but removes the global average pooling layer and directly performs channel dimensionality reduction and increase. Hence, its role is to retain and highlight small target features.

In IVIF tasks, we want networks to recognize and capture the relative importance of decoupled features at different scales. Motivated by the advantages of MS-CAM, we propose an adaptive feature fusion network for IVIF tasks.

## 3. Method

In this section, the adaptive fusion strategy is introduced in Section 3.1, structure information enhancement part is described in Section 3.2, then the overall framework is presented in Section 3.3, and finally the loss functions are described in Section 3.4.

### 3.1. Attention-Based Adaptive Feature Fusion Strategy

The key of feature fusion algorithms is how to explore the relative importance of different features for high-quality fusion. Channel attention mechanism can determine the feature importance of different channels, give different weights to enhance important features and weaken irrelevant features, and thus achieve dynamic selection of important features from infrared and visible images. Previous IVIF methods do not realize this point and employ simple fusion strategies with fixed weights. In this paper, we introduce a multi-scale channel attention mechanism into IVIF tasks and propose an attention-based adaptive fusion (AAF) strategy, which dynamically assigns fusion weights to two input features. As shown in Figure 1a, the multiscale channel attention module *F* generates weight map in channel domain by aggregating the relative importance of global channels and local channels in spatial domain. *F* is formulated by the following equation: (1)F(X)=σ(G(X)⊕L(X)),
where F(X) denotes the generated fusion weight map, G(X) measures the relative importance of features on a global scale, and L(X) measures the relative importance on a local scale. δ(·) denotes the Sigmoid function, and ⊕ denotes the broadcast addition.

Given intermediate features $X\in R^{C\times H\times W}$, Global Average Pooling (GAP) is used to compute the global distribution information $M\in R^{C\times 1\times 1}$ in channel domain. The gating mechanism of two point-by-point convolution layers is adopted to learn the nonlinear interactions of the global distribution information among channels, i.e., 1 × 1 point-by-point convolution with scale factor *r* performs channel dimensionality reduction to obtain $M_{1}\in R^{\frac{C}{r}\times 1\times 1}$, and then channel dimensionality increase is performed by another 1 × 1 convolution to obtain $G\left(X\right)\in R^{C\times 1\times 1}$. The global channel attention features G(X) is formulated in Equation (Equation 2).
(2)$$G\left(X\right)=B(W_{2}\left(\delta \left(B\left(W_{1}\left(g\left(X\right)\right)\right)\right)\right),$$
where $g\left(x\right)=\frac{1}{H\times W}\sum_{i=1}^{H}\sum_{j=1}^{W}X(:,i,j)$ is global average pooling (GAP), δ(·) denotes ReLU activation function, B(·) denotes batch normalization, and $W_{1}$ and $W_{2}$ are point-by-point convolutions with kernel sizes of $1\times 1\times C\times \frac{C}{r}$ and $1\times 1\times \frac{C}{r}\times C$ respectively.

We remove GAP from the global channel attention branch and directly use two point-by-point convolutions to perform channel dimensionality reduction and increase to obtain local channel attention features L(X) with the same size as the input features in order to retain and highlight detail information. $L\left(X\right)\in R^{C\times H\times W}$ is formulated in Equation (Equation 3).
(3)$$L\left(X\right)=B\left(W_{4}\left(\delta \left(B\left(W_{3}\left(X\right)\right)\right)\right)\right),$$
where $W_{3}$ and $W_{4}$ are point-by-point convolutions with kernel sizes of $1\times 1\times C\times \frac{C}{r}$ and $1\times 1\times \frac{C}{r}\times C$, respectively.

As shown in Figure 1b, given two input features $X,Y\in R^{C\times H\times W}$, the AAF strategy based on the multiscale channel attention module *F* can be expressed as

1.Single-layer attention fusion.
(4)$$AAF\left(X,Y\right)=X⊗F\left(X+Y\right)+Y⊗\left(1-F\left(X+Y\right)\right),$$
where ⊗ denotes matrix multiplication. F(X+Y) generates fusion weights consisting of real values from 0 to 1. F(X+Y) and 1−F(X+Y) enable the network to dynamically assign fusion weights between X and Y.2.Multiple iterations of attention fusion.
(5)$$AAF_{n}\left(X,Y\right)=X⊗F\left(AAF_{n-1}\left(X,Y\right)\right)+Y⊗(1-F\left(AAF_{n-1}\left(X,Y\right)\right).$$$AAF_{n}\left(X,Y\right)$ is the fused features generated in *n*-th iteration.

Addition and Concatenation are context-independent strategies. AAF is a fully context-aware soft selection strategy, i.e., the fusion weights are adaptively estimated based on all input features. It is worth noting that there is a performance bottleneck in this fully context-aware approach, i.e., how to synthesize two input features initially. As shown in Equation (Equation 4), employing a simple fusion strategy like X+Y to synthesize *X* and *Y* as input to the AAF module may have an impact on the final fusion weights. Therefore, we use multiple iterations of the AAF module to solve this problem, as shown in Equation (Equation 5). Interactive use of the AAF module improves the input quality to the next AAF module. Considering the marginal effects and the growth of computation demand, we only use a two-layer AAF module $AAF_{2}$ as shown in Figure 1b. In the remaining of this paper, $AAF_{2}$ is denoted as AAF for simplicity. It is noteworthy that the output weights by the AAF module are tensors with the same size as the input features, i.e., each pixel of each feature will be assigned a dynamic weight.

### 3.2. Structure Information Enhancement

Although all current IVIF methods based on feature disentanglement expect the structural features from different sources are as similar as possible, they do differ from each other to a large extent due to the vast discrepancy of different source imaging mechanisms. Different source images provide different dominant structure information, such as the thermal target edge information in infrared images and the rich texture information in visible images. The application of simple fusion strategies such as addition to them is easy to cause weakening of dominant structure information and occurrence of unwanted structure information. In order to provide a complete structure frame that contains all dominant structure information, it is necessary to apply the AAF strategy to the structural features from different sources before reconstruction. Such a structure frame is a prerequisite for super-high quality image fusion. The synthesized structural features *S* under the AAF strategy can be formulated as
(6)$$S=AAF_{2}\left(S_{I},S_{V}\right)=S_{I}⊗F\left(AAF\left(S_{I},S_{V}\right)\right)+S_{V}⊗(1-F\left(AAF\left(S_{I},S_{V}\right)\right),$$
where $S_{I}$ and $S_{V}$ denotes the structural features of infrared and visible images respectively.

### 3.3. Overall Framework

Our aim is to fully synthesize all valuable structure and modal information from different sources and exhibit them in a fused image as more reasonable as possible. The AAF network proposed in this paper is based on an encoder-decoder architecture. The encoder consists of a common convolutional layer Conv1, two structural residual module (SRM) layers, and a convolutional decoupling layer Conv2. The decoupling layer can effectively disentangle structure features from the modal features. We embed SRM in the encoder to aggregate the extracted residual convolutional feature stream with the Sobel gradient information stream for feature reuse, enhancing the network’s ability to describe fine-grained details through the residual edge gradient stream, as shown in Figure 2.

The structure information enhancement part contains an AAF layer. This layer adaptively fuses the structural features extracted from infrared and visible sources into a group of structure features, integrating all of their dominant structure information. The decoder contains three AAF layers and one convolutional layer that enable soft selection of important features from infrared and visible images, allowing the reconstructed images to retain more complete global structural information and meanwhile preserve more modal information. The network structure is shown in Figure 3. To solve the problem of inadequate cross-level fusion, the AAF strategy was used in the cross-level fusion from the lower to the higher layers, preventing detailed information and loss of source images after multiple convolutions and accelerating the training convergence.

In the training phase, we fed the infrared images I and the visible images V into the encoder, decomposed the source images into similar structural features $S_{I}$ and $S_{V}$ with distinct modal information $M_{I}$ and $M_{V}$. To capture all complementary dominant spatial structure information of the source image, we combined $S_{I}$ and $S_{V}$ into the AAF layer to generate the dominant structural features *S*. Then, the modal features of each source $M_{I}$ and $M_{V}$ were fused with *S* through the decoder to reconstruct the source images $\hat{I}$ and $\hat{V}$, respectively. It is noteworthy that we reconstructed the infrared images I and visible images V using the same encoder-decoder architecture. In the test phase, we aimed to fuse all features extracted from both sources. We added a fusion step to the training network by combining the modal features from different sources under addition strategy to generate fused modal features *M*. Here we chose the addition strategy because the extracted modal features from different sources are almost fully complementary. Then, *S* and *M* were fed into the decoder to generate the final fused images. The test network structure is shown in Figure 4.

### 3.4. Loss Function

The loss function consists of three main components, namely image decomposition loss $L_{d}$ and image reconstruction loss $L_{r}$, and global structure loss $L_{\mathrm{structure}}$, the exact definition of the total loss can be expressed as
(7)$$L_{total}\;=L_{d}+L_{r}+L_{structure}.$$

#### 3.4.1. Image Decomposition Loss

The differences between the infrared and visible structural feature maps should be as small as possible. In contrast, the differences between their respective structural and modal information should be as large as possible. Therefore, we extract similar background structure feature maps and unique modal feature maps by calculating their pixel intensity and gradient distribution distances. The image decomposition loss function of the encoder can be defined as
(8)$$\;\;\;\;L_{d}=\;\alpha_{1}\Phi ((||S_{V}-S_{I}{\left|\right|}_{2}^{2})+\alpha_{2}\Phi ((||\nabla S_{V}-\nabla S_{I}{\left|\right|}_{2}^{2})-\alpha_{3}\left(|\right|S_{V}-D_{V}{\left|\right|}_{2}^{2})-\alpha_{4}\left(|\right|S_{I}-D_{I}{\left|\right|}_{2}^{2}),$$
where Φ(·) is the tanh function and ∇ denotes the gradient operator.

#### 3.4.2. Image Reconstruction Loss

In order to drive the pixel intensity, structural similarity index (SSIM) of the reconstructed image as close as possible to the input image. Therefore, the image reconstruction loss function of the decoder is
(9)$$L_{r}=\;\alpha_{5}f\left(I,\hat{I}\right)+\;\alpha_{6}f\left(V,\hat{V}\right),$$
where I and $\hat{I}$, V and $\hat{V}$ represent the input and reconstructed infrared and visible images, respectively.
(10)$$f\left(X,\;\hat{X}\right)\;=||\;X\;-\;\hat{X}\;{\left|\right|}_{2}^{2}\;+\;\lambda L_{SSIM}\left(X,\;\hat{X}\right),$$
where *X* and $\hat{X}$ represent the above input image and reconstructed image, respectively, and λ is a hyperparameter. SSIM can be described as
(11)$$L_{SSIM}\left(X,\;\hat{X}\right)=\;\frac{1\;-SSIM\left(X,\;\hat{X}\right)}{2}.$$

#### 3.4.3. Global Structure Loss

Fully enhancing the dominant structure information in fused images is the core of our network, and thus we designed the global structure loss to achieve this goal. Given a pair of aligned infrared and visible images, we used the Sobel edge filtering operator to generate the edge feature map of the infrared image ∇I and the structure feature map of the visible image ∇V, respectively. The two structure feature maps were synthesized by the “choose-max strategy” to generate a global structure feature map ∇S, which contained the dominant structure information of the source images. The global structure loss proposed in this paper was used to calculate the feature distance between the reconstructed image and ∇S, guiding the AAF module in the structure information enhancement part to integrate all dominant structure information from both sources. The fused image can retain the significant target edge information of the infrared images and the rich texture information of the visible images at the same time.

The global structure feature map can be formulated as
(12)∇S=MAX(∇I,∇V),
where ∇ denotes the gradient filter operator and MAX denotes the maximum fusion rule. Global structure loss $L_{\mathrm{structure}}$ reflects the structural information contained in reconstructed images and can be expressed as
(13)$$L_{structure}=\;\alpha_{7}MSE\left(\nabla \hat{I}-\;\nabla S\right)+\;\alpha_{8}MSE\left(\nabla \hat{V}-\;\nabla S\right).$$

$\nabla \hat{I}$ and $\nabla \hat{V}$ denote the structural feature maps of the reconstructed infrared and visible images, respectively. MSE is the mean-squared error loss. When the difference between the predicted and true values is larger, the penalty of mean-squared loss is larger. Note that $\alpha_{1},\alpha_{2},\alpha_{3},\alpha_{4},\alpha_{5},\alpha_{6},\alpha_{7},\alpha_{8}$ are adjustable hyperparameters.

## 4. Experimental Results and Analysis

In this section, we first describe the experimental details. Then, we present an experimental result comparison of similar models to demonstrate the effectiveness of the AAF model. In addition, we evaluate the fusion performance of the AAF model on two public datasets qualitatively and quantitatively and compare it with five state-of-the-art fusion methods. Finally, several ablation studies were conducted to demonstrate the effectiveness of our specific design.

### 4.1. Experimental Details

To train the proposed fusion model, we selected 180 image pairs from the RoadScene [39] image fusion dataset as the training set. Prior to training, all images in the training set were converted to grayscale maps and cropped centrally with 128 × 128 pixels. In the training phase, the optimizer of the fusion network uses Adam, the learning rate is set to $10^{-3}$, the hyperparameters were set to: $\alpha_{1}=0.3,\alpha_{2}=0.3,\alpha_{3}=0.4,\alpha_{4}=0.4,\alpha_{5}=9,\alpha_{6}=8,\alpha_{7}=5$, $\alpha_{8}=20,\lambda =5$. The fusion model was implemented using Pytorch and experimented on a computer equipped with two NVIDIA 16 G V100 GPUs.

To fully evaluate the generalized performance of the AAF model, we conducted test experiments on the RoadScene [39] and TNO [40] datasets. Thirty-eight TNO image pairs and thirty-six RoadScene image pairs were selected as the test sets. We compare our model with eight existing fusion methods, including three traditional methods, namely IFEVIP [41], HMSD [42], and HMSD_GF [32], four AE-based methods, namely DenseFuse [26] and DIDFuse [20], DRF [17], and Dual-branch Net [19], and one GAN-based method, namely FusionGAN [43]. The code implementations of all these eight methods are publicly available, and during the experiments we adopt the hyperparameter values from the original papers.

The qualitative assessment relies on the subjective evaluation of details such as image texture and contrast by the human visual system, which is not fully convincing. Therefore, we chose six metrics to quantitatively evaluate the fusion results in an objective manner, including entropy (EN) [44], peak signal-to-noise ratio ( PSNR ), visual information fidelity (VIF) [45], spatial frequency (SF), standard deviation (SD), and mean gradient (MG). EN measures the amount of information contained in the fused image from an information-theoretic perspective. PSNR measures the fusion performance in terms of how similar the fused image is to the source image, with larger values indicating less distortion in the fusion process. VIF measures the information fidelity of the fused image from the perspective of the human visual system by calculating the distortion of the image. SF measures the spatial frequency information contained in the fused image, and a larger SF indicates that the fused result contains more texture edge information. MG reflects the texture and structure information of the image, and SD reflects the contrast of the fused image from a statistical point of view. Fusion algorithms with larger EN, PSNR, VIF, SF, SD and MG indicate a better fusion performance.

### 4.2. Experimental Comparison of Similar Models

The proposed AAF fusion method is a representation disentanglement fusion model. The biggest difference between our method and other representation disentanglement-based methods is that we adopted an adaptive fusion strategy for the synthesis of structural features from different sources before reconstruction and the synthesis of modal features with structural features, while simple fusion strategies are adopted in other similar methods. In order to demonstrate the effectiveness of our method, we choose other three representation disentanglement-based fusion methods, DIDFuse [20], DRF [17] and Dual-branch Net [19], for comparison.

**Qualitative evaluation**. As shown in Figure 5, we selected some representative regions in the source and fused images and zoomed in to view them. We can see that, for images containing pedestrian, other methods suffer from poor target saliency, low contrast, blurred edges, and inconspicuous high-frequency details. Similarly, for images of natural landscapes or street scenes, other methods suffer from false gray skies, poor sharpness, blurred edge contours of tree branches, and low color contrast. In contrast, our method is able to obtain fused images with high contrast, brighter targets, and more texture detail information retained.

**Quantitative evaluation**. From Table 2 and Table 3, we can see that our EN is always higher than that of other methods, indicating that our fusion strategy retains more information. Our SF and MG are much higher than those of other methods, indicating that our model has a clear advantage in texture detail and edge information preservation.In terms of SD values, our model achieves the maximum optimal, indicating that the fused images by our model provide highest contrast. The AAF fusion strategy adopted by our model is a channel attention mechanism, which can determine the feature importance of different channels, give different weights to enhance important features and weaken irrelevant features. As a result, the error between the pixel value of the generated fusion image and the single source image increases, which affects the performance of PSNR index. In addition, the VIF algorithm compares each image region of the source image and the fusion image equally from the perspective of the human visual system. The attention mechanism may influence the final assessment. However, our model still achieves high levels in terms of VIF and PSNR values, indicating that our model’s fused images provide excellent low distortion results.

### 4.3. Fusion Performance Comparison Test

It is well known that generalization performance is an important aspect for evaluating deep learning-based methods. Therefore, we performed generalization experiments on the TNO and RoadScene datasets to demonstrate the generalization performance of the proposed AAF model.

Qualitativeevaluation.Figure 6 shows the qualitative comparison results of AAF methods with five other state-of-the-art methods on the TNO and RoadScene test datasets. We selected three and two pairs of typical infrared and visible image pairs from the TNO and RoadScene test datasets, respectively. All images were converted to grayscale, and the image pairs were pre-aligned and had the same resolution. We show the source images in the first two top rows of Figure 6, followed by the fused images by IFEVIP, HMSD, HMSD_GF, FusionGAN, DenseFuse, and our AAF model. The three columns on the left: the TNO test dataset, and the two columns on the right: the RoadScene test dataset. We selected some representative regions in the source and fused images zoomed in to view them. Observing the details of the fused images, such as the sky and the houses in the first column, and so on, we can find that the texture information of the background in the fused images by other methods is disturbed by the thermal radiation information. iFEVIP, HMSD, HMSD_ GF, FusionGAN, DenseFuse cannot reflect the real sky, background, and other targets. Our fusion results are closer to the real sky, i.e., the advantageous information provided by the visible source. While the images generated by IFEVIP and FusionGAN methods retain the significant target information of infrared source, rarely retain enough detail information of visible images. The images generated by DenseFuse method have blurred edges and low contrast. The images generated by the HMSD and HMSD_ GF methods are worse than those by our method in terms of brightness and contrast. The fused images generated by our method can reflect all advantageous information from both sources, a more complete structure frame of the original scene with richer texture details, more significant infrared target information, and higher contrast.

Quantitativeevaluation. The comparative results of different methods on the six metrics are shown in Figure 7 and Figure 8, Table 4 and Table 5. Obviously, our method obtains the best scores on EN, SF, MG, SD, and VIF, except PSNR. The AAF fusion strategy increases the error between the fused image and the single original image in pixel value, which reduces the performance of PSNR index. However, PSNR is not a perfect image quality evaluation index; it cannot fully reflect the difference in image fusion quality and should be combined with other image quality evaluation indicators for comprehensive evaluation.The best EN, SF, and MG indicate that our method generates fused images with more information, higher resolution, and clearer textures. The best performance on SD and VIF metrics indicate that our algorithm generates fused images with better visual effects. The quantitative evaluation results verifies the obvious effectiveness of adaptive fusion strategy in IVIF tasks.

### 4.4. Ablation Experiments

#### 4.4.1. Edge and Texture Retention Analysis

Our model retains the dominant structural information from both sources and relies partially on the global structure loss to guide the AAF module in the structure information enhancement part, adaptively fusing the thermal target edge information from infrared source and the rich texture information from visible source. To verify the importance of the global structure loss and the AAF module in the structure information enhancement part, we performed two ablation experiments by removing one of them, while keeping the remaining network structure unchanged. More specifically, we trained our fusion model without global structural loss and replacing the structure information enhancement part by concatenation, called NO_Loss and NO_S_AAF, respectively.

Some typical examples are shown in Figure 9. Carefully observing the zoomed-in regions of the red-framed bushes in the first column and the road sign in the fourth column, we find that the model with the structure information enhancement part generates fused images with higher target saliency (from infrared source) and clearer texture details (from visible source) than the model without the enhancement part. The zoomed-in regions of green-framed utility poles in the third column and green-framed tires in the fifth column of Figure 9 exhibit very smooth texture details and blurred edges, respectively, for the model without the global structure loss. In contrast, the model with the global structure loss maintains the visible texture information and the high frequency salient infrared information. We also did quantitative ablation analysis on TNO and RoadScene datasets. As shown in Table 6 and Table 7, SF and MG metrics of NO_Loss and NO_S_AAF are much lower than those of the full model, undoubtedly verifying that both the global structure loss and the structure information enhancement part better preserve the thermal target edge information from infrared source and rich texture information from visible source.

#### 4.4.2. AAF Fusion Strategy Ablation Analysis

To verify the necessity of the adaptive fusion strategy in improving the fusion quality of fusion, we performed ablation experiments, while keeping the remaining network structure and loss function unchanged. We trained a model that only adopted a concatenation fusion strategy to replace all AAF modules in the network. We refer to this method as S_CAT. We selected two pairs of images from each of the TNO and RoadScene test datasets for qualitative comparison. As shown in Figure 10, the model with concatenation strategy generates fused images with blurred edge contours, little high-frequency detail, and unnecessary gradient variations around the highlighted targets. In contrast, the model with the AAF module generates fused images with salient targets, sharper edge textures, higher contrast, and better visual effects. Table 8 and Table 9 show the six metrics of the fusion results for TNO and RoadScene test datasets, respectively. The model with the AAF module has the best performance on all six metrics, which demonstrates the superior performance of adaptive strategy relative to concatenation strategy.

### 4.5. Efficiency Comparison

To analyze the complexity of the AAF model, we tested the average running time on the TNO and RoadScene datasets, as shown in Table 10. All traditional methods were run on the CPU, while deep learning methods DenseFuse, FusionGAN, and OURS were run on the GPU. Since deep learning methods include training and testing, we only recorded the testing time for comparison. The results in the table show that our AAF model has moderate time complexity, which can effectively meet the task requirements while maintaining good performance.

## 5. Conclusions

To address the problems that previous IVIF methods ignore—the complementarity of the scene structure features from different sources and the limitation of simple fusion strategies—this paper proposes an attention-based adaptive fusion network. The attention-based adaptive fusion strategy is able to measure the relative importance of features and adaptively assign dynamic weights to different features. We adopted this adaptive strategy to enhance dominant structure information from both sources before reconstruction and fused the structure features and modal features into reconstructed images. A global structure loss function was also proposed to guide the structure enhancement architecture to retain all dominant source information. Based on extensive qualitative and quantitative experiments, our method can fully fuse the target edge information from the infrared source and the rich texture information from a visible source into one image with high resolution, high contrast, and excellent visual effect. 

## Figures and Tables

**Figure 1 entropy-25-00407-f001:**
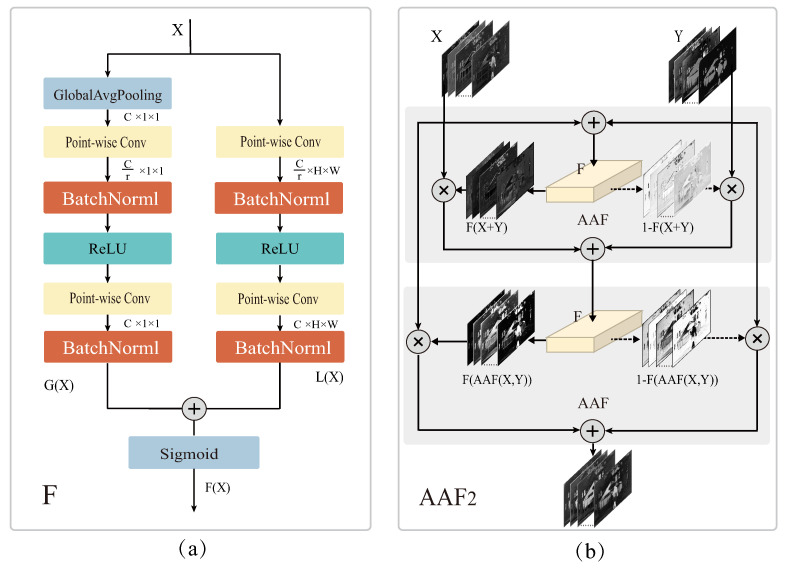
Schematic of AAF strategy. (**a**) The multiscale channel attention module *F*. (**b**) The two-layer iterative AAF module.

**Figure 2 entropy-25-00407-f002:**
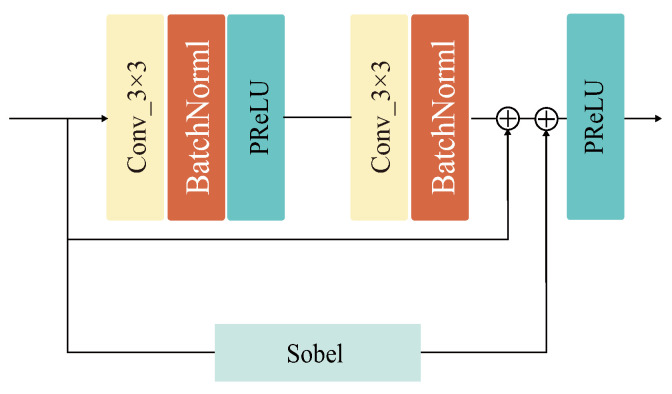
Specific design of the structural residual module (SRM). Structural residual parallel Sobel operator consisting of two 3 × 3 convolution kernels, two batch normalization layer, and two PReLU activation function.

**Figure 3 entropy-25-00407-f003:**
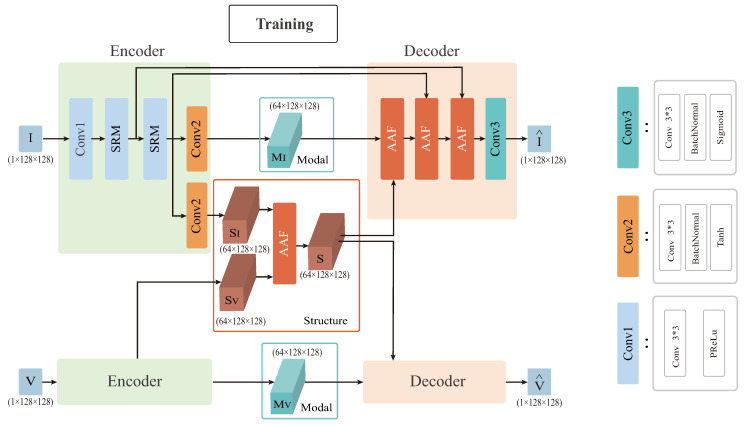
Schematic diagram of the overall AAF framework. SRM is the structural residual module and AAF is attention-based adaptive fusion module. conv1 consists of a 3 × 3 convolution kernel, a PReLU activation function, conv2 consists of a 3 × 3 convolution kernel, a batch normalization layer, a Tanh activation function, and conv3 consists of a 3 × 3 convolution kernel, a batch normalization layer, a Sigmoid activation function.

**Figure 4 entropy-25-00407-f004:**
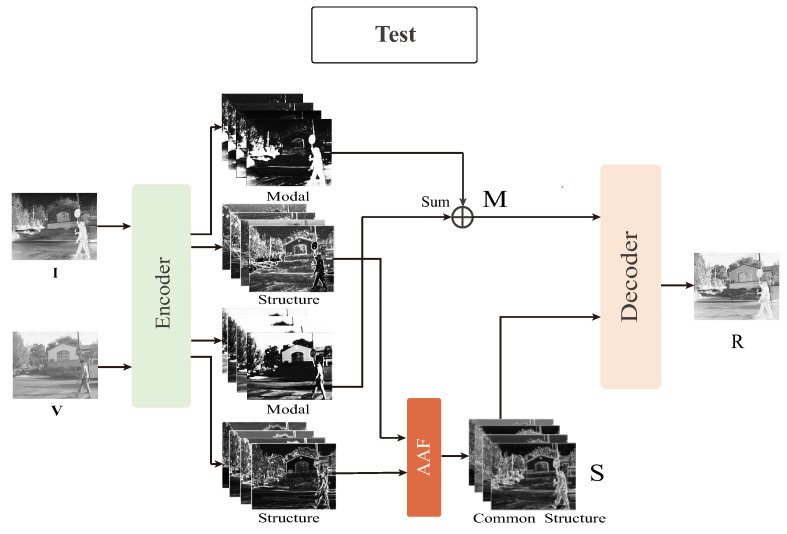
Schematic diagram of AAF test framework. *M* represents the added modal features and *R* is the fused image.

**Figure 5 entropy-25-00407-f005:**
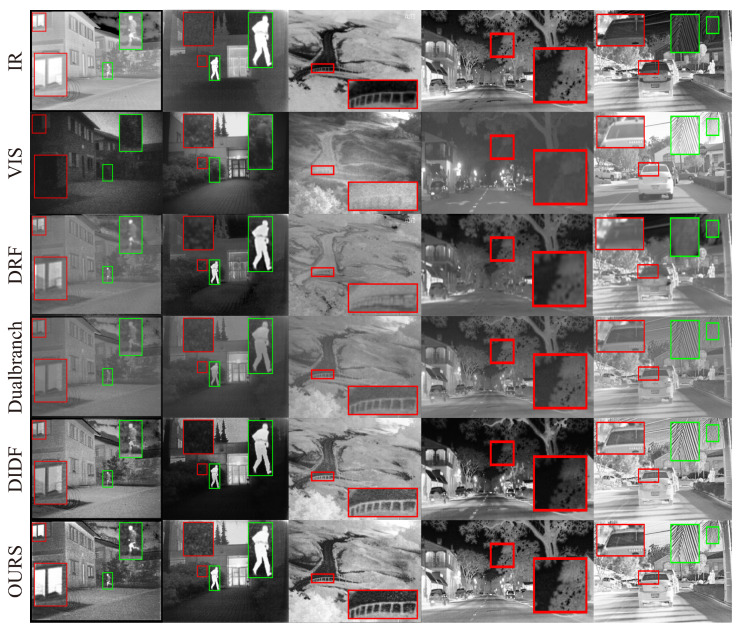
Qualitative experimental result comparison of the AAF model with DRF, Dual-branch Net and DIDFuse on the TNO and RoadScene datasets. The first two rows are infrared and visible images, and the following are the fused images by DRF, Dualbranch, DIDFuse, and our AAF model in order. Three columns on the left: TNO dataset, two columns on the right: RoadScene dataset.

**Figure 6 entropy-25-00407-f006:**
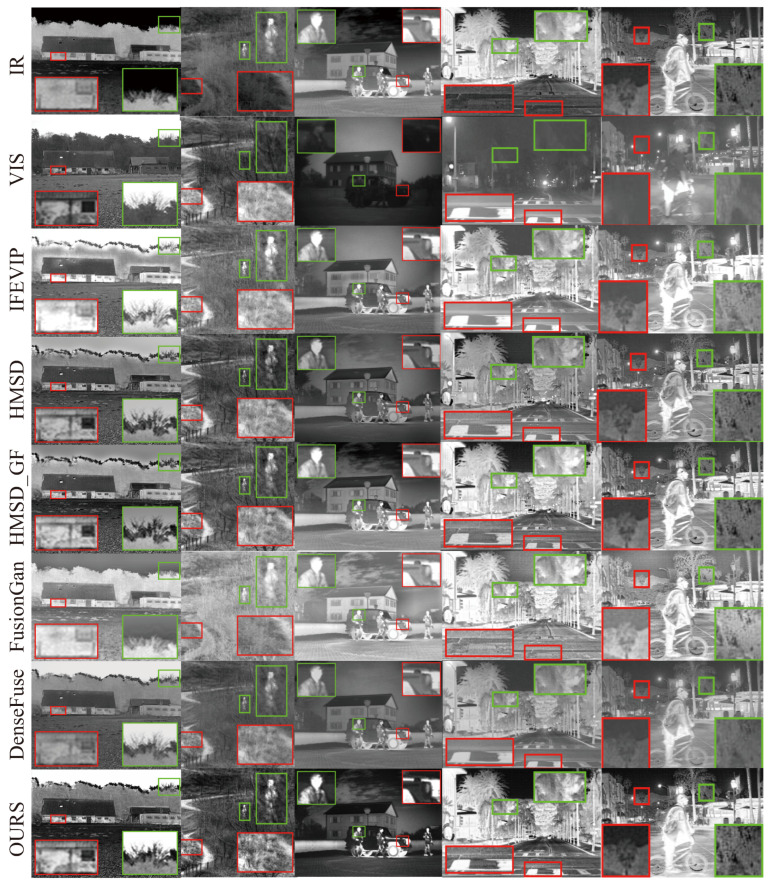
Qualitative comparison of AAF model with five state-of-the-art IVIF methods on the TNO and RoadScene test datasets. The first two top rows are the infrared and visible images, and the following are the fused images by IFEVIP, HMSD, HMSD_GF, FusionGAN, DenseFuse, and our AAF model. The three columns on the left: the TNO dataset, and the two columns on the right: the RoadScene dataset.

**Figure 7 entropy-25-00407-f007:**
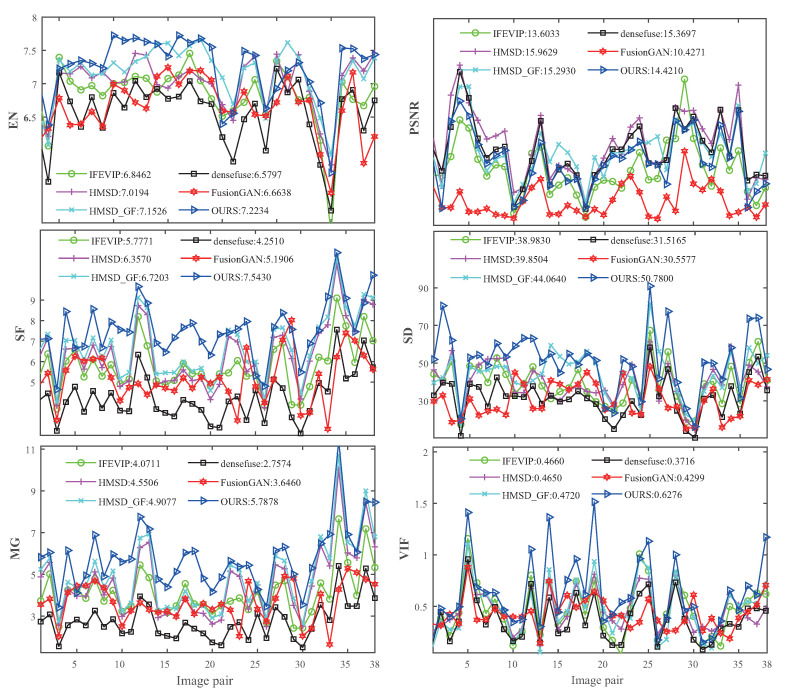
Quantitative comparison results of the AAF model with five state-of-the-art methods on six metrics for the TNO test dataset.

**Figure 8 entropy-25-00407-f008:**
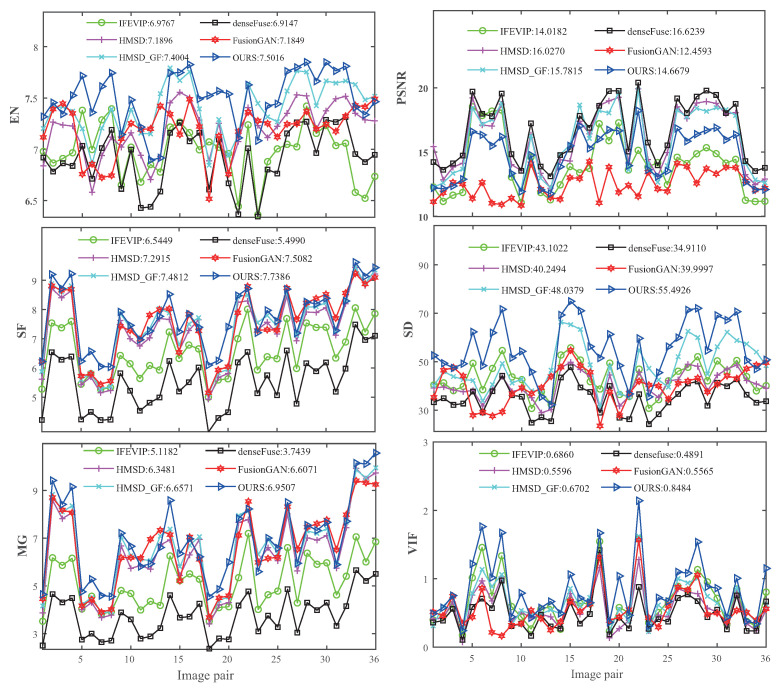
Quantitative comparison results of the AAF model with five state-of-the-art methods on six metrics for the RoadScene test dataset.

**Figure 9 entropy-25-00407-f009:**
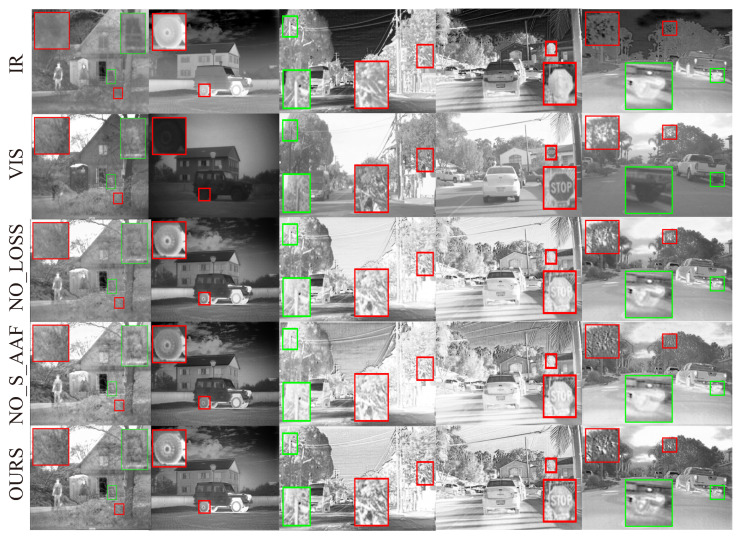
The two columns on the left: TNO test dataset, and the three columns on the right: RoadScene test dataset. The first two top rows are the source images, followed by the fused images by the NO_Loss model and NO_S_AAF model and the full model, respectively.

**Figure 10 entropy-25-00407-f010:**
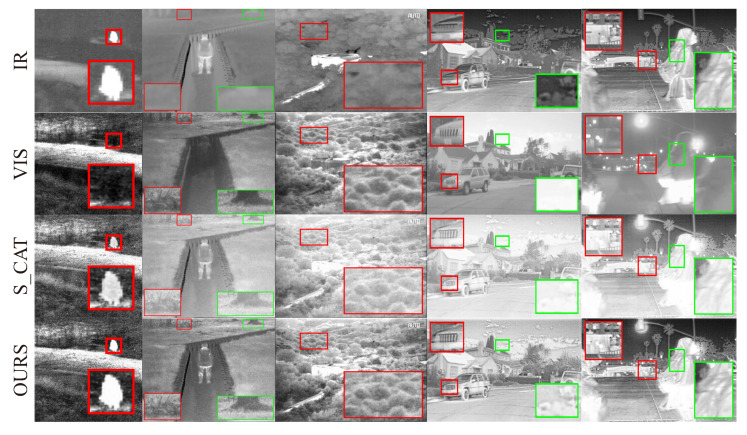
Qualitative ablation study of AAF strategy. The three columns on the left: TNO test dataset and the two columns on the right: RoadScene test dataset. The first two top rows are the source images, and the following are the fusion images by the model with concatenation strategy and those by the AAF model.

**Table 1 entropy-25-00407-t001:** Feature fusion formulas for different fusion strategies. G(·) denotes the weight generation module and ⊗ denotes the matrix multiplication.

Context-Aware	Type	Formulation	Example
	Addition	X + Y	DRF [17], Dual-branch [19]
	Concatenation	$W_{A}X_{:,i,j}+W_{B}Y_{:,i,j}$	DIDFuse [20], CUFD [18]
None	Average	$\frac{1}{2}X+\frac{1}{2}Y$	TIF [33]
	Choose-max	$X_{:,i,j}/Y_{:,i,j}$	GFF [11], IFCNN [24]
	Max-$l_{1}$	$W_{A\max}X+W_{B\max}Y$	NGDC [13]
	$L_{1}$-norm	$W_{A}X+W_{B}Y$	DenseFuse [26]
Fully	Soft Selection	G(X + Y) ⊗ X + (1 − G(X + Y)) ⊗ Y	SKNet [34]

**Table 2 entropy-25-00407-t002:** Quantitative comparison of AAF model with DRF, Dual-branch Net and DIDFuse on the TNO dataset. The best, second best and third best values are indicated in black bold, red and blue, respectively.

Methods	EN	SF	MG	PSNR	SD	VIF
DRF	6.4773	3.0594	2.1140	14.0305	28.2402	0.3480
Dualbranch	6.3507	3.5606	2.3751	**15.5899**	24.4902	0.2988
DIDF	7.1002	6.0910	4.3534	13.9378	48.0636	**0.6456**
OURS	**7.2234**	**7.5430**	**5.7878**	14.4210	**50.7800**	0.6276

**Table 3 entropy-25-00407-t003:** Quantitative comparison of AAF model with DRF, Dual-branch Net and DIDFuse on the RoadScene dataset. The best, second best and third best values are shown in black bold, red and blue, respectively.

Methods	EN	SF	MG	PSNR	SD	VIF
DRF	7.2503	4.7228	3.4275	14.3983	44.6414	0.5403
Dualbranch	6.7988	4.9488	3.3732	**16.4648**	31.0218	0.4433
DIDF	7.3795	6.8482	5.6517	14.8007	52.0672	0.7935
OURS	**7.5016**	**7.7386**	**6.9507**	14.6679	**55.4926**	**0.8484**

**Table 4 entropy-25-00407-t004:** Quantitative comparison of AAF model with five state-of-the-art methods on six metrics for the TNO test dataset, with the best, second-best and third-best values in black bold, red and blue, respectively.

Methods	EN	SF	MG	PSNR	SD	VIF
IFEVIP	6.8462	5.7771	4.0711	13.6033	38.9830	0.4660
HMSD	7.1094	6.3570	4.5506	**15.9629**	39.8504	0.4650
HMSD_GF	7.1526	6.7203	4.9077	15.2930	44.0640	0.4720
DenseFuse	6.5797	4.2510	2.7574	15.3697	31.5165	0.3716
FusionGAN	6.6638	5.1906	3.6460	10.4271	30.5577	0.4299
OURS	**7.2234**	**7.5430**	**5.7878**	14.4210	**50.7800**	**0.6276**

**Table 5 entropy-25-00407-t005:** Quantitative comparison of AAF model with five state-of-the-art methods on six metrics for the RoadScene test dataset, with the best, second-best and third-best values in black bold, red and blue, respectively.

Methods	EN	SF	MG	PSNR	SD	VIF
IFEVIP	6.9767	6.5449	5.1182	14.0182	43.1022	0.6860
HMSD	7.1896	7.2915	6.3481	16.0270	40.2494	0.5596
HMSD_GF	7.4004	7.4812	6.6571	15.7815	48.0379	0.6702
DenseFuse	6.9147	5.4990	3.7439	**16.6239**	34.9110	0.4891
FusionGAN	7.1849	7.5082	6.6071	12.4593	39.9997	0.5565
OURS	**7.5016**	**7.7386**	**6.9507**	14.6679	**55.4926**	**0.8484**

**Table 6 entropy-25-00407-t006:** Six metrics of the fused images by the NO_Loss model and NO_S_AAF model and the full model on TNO test dataset, respectively. The best values are in bold.

Methods	EN	SF	MG	PSNR	SD	VIF
NO_Loss	7.1109	6.4755	4.6123	12.3098	50.4393	0.6457
NO_S_AAF	**7.2530**	6.9701	5.2198	14.1635	49.9152	**0.6470**
OURS	7.2234	**7.5430**	**5.7878**	**14.4210**	**50.7800**	0.6276

**Table 7 entropy-25-00407-t007:** Six metrics of the fused images by the NO_Loss model and NO_S_AAF model and the full model on RoadScene test dataset, respectively. The best values are in bold.

Methods	EN	SF	MG	PSNR	SD	VIF
NO_Loss	7.3880	6.8765	5.7245	**14.689**	54.2375	0.8454
NO_S_AAF	7.4297	6.6505	6.0720	13.7379	54.8860	**0.8520**
OURS	**7.5016**	**7.7386**	**6.9507**	14.6697	**55.4926**	0.8484

**Table 8 entropy-25-00407-t008:** Six metrics of the fused images by the concatenation model and the AAF model on TNO test dataset, respectively. The best values are in bold.

Methods	EN	SF	MG	PSNR	SD	VIF
S_CAT	7.1671	6.5785	4.8097	13.1636	45.6796	0.5864
OURS	**7.2234**	**7.5430**	**5.7878**	**14.4210**	**50.7800**	**0.6276**

**Table 9 entropy-25-00407-t009:** Six metrics of the fused images by the concatenation model and the AAF model on RoadScene test dataset, respectively. The best values are in bold.

Methods	EN	SF	MG	PSNR	SD	VIF
S_CAT	7.0836	6.3428	4.9261	12.3407	44.0769	0.6438
OURS	**7.5016**	**7.7386**	**6.9507**	**14.6679**	**55.4926**	**0.8484**

**Table 10 entropy-25-00407-t010:** Average running time of different methods on two datasets (unit: second).

Datasets	IFEVIP	HMSD	HMSD_GF	DenseFuse	FusionGAN	OURS
TNO	0.034	3.224	0.644	0.056	0.224	0.265
RoadScene	0.029	1.555	0.317	0.046	0.119	0.193

## Data Availability

The data presented in this study are available on request from the corresponding author.

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
