# Peer review of "Infrared and Visible Image Fusion via Attention-Based Adaptive Feature Fusion"

_entropy, 2023, doi:10.3390/e25030407_

Round 1
Reviewer 1 Report
The authors designed an adaptive fusion network to synthesize decoupled common structural features and distinct modal features under an attention-based adaptive fusion (AAF) strategy. The structure of the whole paper is quite complete. But I have the following add as questions that should as follows.
1. Is it possible to use your "Attention-based Adaptive Feature Fusion" in the real-time image (video)? If possible, how to use it in a real case?
2. In Table 2, please describe why PSNR and VIF are not the best ones compared with DRF, Dualbranch, DIDF, and the proposed method in more detail.
3. In Table 3, please describe why PSNR is not the best one compared with DRF, Dualbranch, DIDF, and the proposed method in more detail.
4. In Table 4, please describe why PSNR is not the best one compared with IFEVIP, HMSD, HMSD_GF, DenseFuse, FusionGAN, and the proposed method in more detail.
5. In Table 5, please describe why PSNR is not the best one compared with IFEVIP, HMSD, HMSD_GF, DenseFuse, FusionGAN, and the proposed method in more detail.
6. In Table 6, please describe why EN and VIF are not the best ones compared with NO_Loss, NO_S_AAF, and the proposed method in more detail.
7. In Table 7, please describe why PSNR and VIF are not the best ones compared with NO_Loss, NO_S_AAF, and the proposed method in more detail.
8. Please list all computational time of fused images generated by different models, such as IFEVIP, HMSD, HMSD_GF, FusionGAN, DenseFuse, and the proposed AAF model in this paper.
Author Response
Dear reviewer:
Thank you for your decision and constructive comments on my manuscript. We have seriously considered the questions raised by Reviewer and made some modifications and replies.
Please see the attachment.
Sincerely,
Lei Wang

Reviewer 2 Report
This paper proposes a novel adaptive fusion network in order to cope with the infrared and visible image fusion problem. The main contributions are the design of the attention-based adaptive fusion strategy and the adaptive fusion network. This topic is interesting and suitable for the transactions. Qualitative and quantitative results are shown in order to prove the superiority of the proposed method. The paper is clearly written and well organised. Qualitative and quantitative experimental results are convincing. Overall, it is a good work in my view and worth publishing as a paper. I have a bit of comments.
My detailed comments are as follows:
1. Details on the complexity of the proposed adaptive fusion network should be included.
2. A comparison of the running time of the proposed and the compared methods should also be included.
Author Response

(The authors gave the same response as above.)
